# Performance Evaluation of a New Sport Watch in Sleep Tracking: A Comparison against Overnight Polysomnography in Young Adults

**DOI:** 10.3390/s24072218

**Published:** 2024-03-30

**Authors:** Andrée-Anne Parent, Veronica Guadagni, Jean M. Rawling, Marc J. Poulin

**Affiliations:** 1Department of Physiology & Pharmacology, Cumming School of Medicine, University of Calgary, Calgary, AB T2N 1N4, Canada; andree-anne_parent@uqar.ca (A.-A.P.); veronica.guadagni@cerebrahealth.com (V.G.); 2Hotchkiss Brain Institute, Cumming School of Medicine, University of Calgary, Calgary, AB T2N 1N4, Canada; 3Libin Cardiovascular Institute of Alberta, Cumming School of Medicine, University of Calgary, Calgary, AB T2N 1N4, Canada; 4Department of Family Medicine, Cumming School of Medicine, University of Calgary, Calgary, AB T2N 1N4, Canada; jmrawlin@ucalgary.ca; 5Department of Clinical Neurosciences, Cumming School of Medicine, University of Calgary, Calgary, AB T2N 1N4, Canada; 6Faculty of Kinesiology, University of Calgary, Calgary, AB T2N 1N4, Canada; 7O’Brien Institute of Public Health, Cumming School of Medicine, University of Calgary, Calgary, AB T2N 1N4, Canada

**Keywords:** agreement, sleep stages, sleep tracker, exercise, accelerometry, photoplethysmography

## Abstract

**Introduction**: This study aimed to validate the ability of a prototype sport watch (Polar Electro Oy, FI) to recognize wake and sleep states in two trials with and without an interval training session (IT) 6 h prior to bedtime. **Methods**: Thirty-six participants completed this study. Participants performed a maximal aerobic test and three polysomnography (PSG) assessments. The first night served as a device familiarization night and to screen for sleep apnea. The second and third in-home PSG assessments were counterbalanced with/without IT. Accuracy and agreement in detecting sleep stages were calculated between PSG and the prototype. **Results**: Accuracy for the different sleep stages (REM, N1 and N2, N3, and awake) as a true positive for the nights without exercise was 84 ± 5%, 64 ± 6%, 81 ± 6%, and 91 ± 6%, respectively, and for the nights with exercise was 83 ± 7%, 63 ± 8%, 80 ± 7%, and 92 ± 6%, respectively. The agreement for the sleep night without exercise was 60.1 ± 8.1%, k = 0.39 ± 0.1, and with exercise was 59.2 ± 9.8%, k = 0.36 ± 0.1. No significant differences were observed between nights or between the sexes. **Conclusion**: The prototype showed better or similar accuracy and agreement to wrist-worn consumer products on the market for the detection of sleep stages with healthy adults. However, further investigations will need to be conducted with other populations.

## 1. Introduction

Sleep deprivation can negatively impact decision making, increase anxiety and irritability, and elevate the risk of cardiovascular and gastrointestinal diseases, mental illness, and cancer [1]. Moreover, sleep plays a key role in cognitive function and physical recovery that can affect academic performance and sports performance [1,2,3,4,5]. Objective sleep quality measurements derived from wearables could provide an alternative opportunity to the use of sleep diaries to analyze individuals’ sleep patterns for a prolonged period of time and suggest changes to sleep hygiene and the implementation of behavioral interventions that could prevent the negative health consequences of poor sleep [6,7]. Sleep quality can be measured in a variety of ways that complement each other. The gold standard assessment to evaluate objective sleep quality is the analysis of signals obtained from overnight polysomnography (PSG), which monitors several physiologic variables over a full night including sleep stages (non-rapid eye movement (NREM) and rapid eye movement (REM) sleep) breathing, oxygenation, leg movements, and body position [8]. The current technology allows PSG to be performed in the individual’s home, rather than in a sleep lab, which makes it more cost-effective and has better ecological validity. However, the majority of home PSG devices require a specialist to set up the equipment and to score the data. To minimize costs, effort, and invasion of individuals’ privacy, it is attractive to use tools such as subjective questionnaires together with wearables to assess sleep quality. Questionnaires such as the Pittsburgh Sleep Quality Index (PSQI) [9] provide a subjective appraisal of sleep quality for the month prior to the questionnaire’s administration.

The term wearables comprises devices worn on the body which provide physiological measures and/or feedback [10,11]. Different commercial sleep tracking wearables have become available during the last decade (Table 1). For example, wrist actigraphy utilizes an accelerometer built in a wristwatch, and it records movement as a surrogate measurement of wakefulness and inactivity as a proxy measurement of sleep [3,12]. The majority of first-generation sleep trackers included a tri-axial accelerometer to use movement data to determine sleep/awake timing [7]. Measurements collected by these devices included total sleep time (TST), total time scored as sleep, sleep onset latency (SOL, the time from when the participant goes to bed (lights out) to the first epoch of sleep), wake after sleep onset (WASO), and sleep period time (SPT) also called Time in Bed (TIB), which refers to the time a participant spends lying in bed [3,13,14,15]. Detection of these sleep parameters with actigraphy has been shown to have a moderate correlation with PSG, often overestimating TST [16]. In addition, some applications on smartphones have been developed which use either accelerometry and/or light sensors and/or a microphone to detect sleep/awake states [17,18]. Nonetheless, using a smartphone application to quantify sleep has many limitations that impact the validity, such as the location of the sleep tracker during the night and the interference from a partner [17].

Other wearables also utilize various physiological signals to measure continuous objective parameters of sleep and wakefulness over 24 h that can be collected for extended periods of time [3,10,19,20]. In addition to recording the individual’s sleep schedule, data on sleep fragmentation can be obtained from these unobtrusive recording devices. Furthermore, the use of heart rate variability (HRV) to derive sleep architecture has been proposed as a possible alternative to the direct measurement of electroencephalography (EEG) to detect sleep stages and could be conducted using heart rate data from wearables [6]. However, HRV is sensitive to many factors such as exercise practice and an individual’s sex [21,22]. Notably, exercise prior to bedtime was reported to influence HRV during sleep in a previous study, which could then impact the validity of the prototypes that use this signal [23]. Together, these real-world data collected with user-friendly wearables can help a coach or health professional; for example, to assess athletes’ sleep quality for many nights during the training period and propose, if needed, sleep hygiene recommendations [5]. However, the fact remains that the validity of these devices is still debatable, leading to controversy in their use for clinical and research purposes [24,25].

The aim of the current study is to evaluate the performance of a new sport watch (Polar Electro Oy, Finland), with a triaxial wrist accelerometry sensor and a photoplethysmography sensor, and to compare the ability of its HRV algorithm to detect sleep stages against the gold standard, i.e., PSG. The target measurements in this comparison are sleep stages: rapid eye movement (REM), light sleep (stage 1 (N1) and stage 2 (N2)), and deep sleep (N3). The detection of sleep/wake states was performed during a night of sleep without exercise during the evening and during a night of sleep preceded by interval training during the evening. Here we hypothesize that the following: (1) the ability of the new product to differentiate sleep stages will be similar to that of other wrist sleep trackers on the market which use accelerometry and photoplethysmography at the wrist [25], and (2) the timing of exercise will impact the validity of the measurements due to its influence on HRV [21,22]. The second hypothesis is of particular relevance because the target market for sport watch wearables is an active population.

## 2. Materials and Methods

### 2.1. Participants

Healthy volunteers were recruited in the city of Calgary, Canada through poster and email notifications from the Cumming School of Medicine at the University of Calgary and Foothills Medical Centre. The eligibility of interested participants was assessed over the telephone with a screening questionnaire. The inclusion criteria included: age between 18 and 35 years, body mass index (BMI) ≤ 30 kg/m, residence in Calgary > 1 year, no prior diagnosis of obstructive sleep apnea (OSA), no current medications known to affect vasculature, and no history of cardiorespiratory disease, hypertension, or smoking. A total of 15 participants were not eligible due to these inclusion criteria. Participants and their physicians provided written informed consent. This study was approved by the University of Calgary Institutional Conjoint Health Research Ethics Board (Protocol # REB16-1027). Before the first fitness assessment, 10 participants withdrew from this study due to time constraints. After the familiarization study night, one participant was excluded due to obstructive sleep apnea (OSA), and their family doctor was notified. Figure 1 illustrates the flow of participants through every step of this study.

### 2.2. Measurements and Procedures

This study included a fitness assessment, in-home polysomnography familiarization night, in-home polysomnography without exercise, and in-home polysomnography with an interval training exercise session less than 6 h prior to participants’ bedtime. The subsequent sections outline details of the different assessments and obtained measurements.

### 2.3. Assessment of Physical Fitness

The physical fitness assessment was conducted in the Clinical and Translational Exercise Physiology (CTEP) laboratory, Cumming School of Medicine, University of Calgary by certified exercise physiologists (Canadian Society of Exercise Physiology) and included anthropometric measurements, assessment of grip strength, and measurement of maximal aerobic capacity ($\dot{V}$O_2_ peak). Before the assessment, participants were instructed, according to the Canadian Society for Exercise Physiology (CSEP) guidelines [26], not to perform the following: (1) eat a heavy meal four hours prior to the assessment, (2) perform vigorous exercise 24 h prior to the assessment, or (3) ingest caffeine and/or alcohol six hours prior to the assessment. The anthropometric measurements consisted of weight and height (Healthometer, Sunbeam, Boca Raton, FL, USA), bioelectrical impedance (Quantum IV, RJL Systems, Clinton Township, MI, USA), 7-site skin fold [27,28], and waist and hip circumference according to CSEP guidelines [26]. The grip strength (78,010 Hand dynamometer, Lafayette Instrument, Lafayette, IN, USA) was taken twice for both hands [26]. Participants rested for at least five minutes prior to the resting heart rate and blood pressure measurements in preparation for the $\dot{V}$O_2_ peak test. A motorized treadmill (TMX428CP Trackmaster, Full Vision Inc., Newton, KS, USA) was used to conduct a modified Bruce protocol [29] to assess the $\dot{V}$O_2_ peak. The following outcome measurements were obtained by a metabolic analyzer (Vmax Encore 29, CareFusion, San Diego, CA, USA) and a 12-lead ECG (CardioSoft, GE Healthcare, Chicago, IL, USA): oxygen uptake, ventilation, carbon dioxide production, and heart rate. The participant was asked about his/her perceived exertion every 3 stages with a Borg scale. The test stopped when the participant was exhausted or unable to sustain the exercise. The participant was released when his/her heart rate was under 100 beats/min and his/her blood pressure was under 140/90 mmHg.

### 2.4. In-Home Polysomnography

In-home, unattended polysomnography (PSG) was conducted three times; the first time was used to familiarize the participant with the equipment and to screen for possible undiagnosed OSA, while the second and third nights included the use of the prototype sport watch (Polar Electro Oy, Kempele, Finland) with the commercially available Sleep Plus Stages sleep tracking algorithm. The protocol for the two nights with the prototype sport watch included a night with an interval exercise session during the evening before the participant’s bedtime (<6 h prior to bedtime) and another night without any exercise 7 h prior to the participant’s bedtime. The order of the two nights was counterbalanced, and the sleep nights needed to be completed within a 14-day window. Trained technicians went to the participant’s home to set up the polysomnography equipment (Embletta MPR PG and ST unit, Natus Neurology Inc., Middleton, WI, USA) a few hours before the participant’s bedtime. The prototype consisted of photoplethysmography and accelerometry built into a wrist band that fit tightly on the left forearm behind the processus styloideus ulna, in accordance with the device manufacturer’s guidelines. The PSG measurements included respiratory parameters collected through finger pulse oximetry, thoracic and abdominal inductive plethysmography, nasal cannula for pressure, and an oronasal thermistor. As well, leg movements were measured using 2 electromyography (EMG) electrodes placed on each anterior tibialis muscle vertically separated by 2–4 cm, and the eye and mandible movement was recorded with 2 EMG electrodes on the rectus muscles and 3 EMG electrodes on the depressor muscles. The body position was derived by 3D gravity sensors in the Embletta unit (256 Hz). Finally, 10 EEG electrodes (including F3, F4, FZ, CZ, C3, C4, O1, O2, M1, M2) were placed on the skull and mastoid bones following the standard 10–20 system electrode placement. To ensure time synchronization, both PSG and the sport watch were connected at the same time. All recordings were performed at the participant’s home, where the sound and temperature were not controlled (i.e., free-living monitoring environment). The data were scored by a sleep technician using a semi-automatic sleep scoring system (MICHELE Sleep Scoring System, Cerebra Health, Winnipeg, MB, Canada) to determine sleep stages scored in 30-s epochs, sleep period time, and total sleep time [19,20]. During familiarization, PSG data were analyzed to recognize potential sleep disorders (exclusion criteria) according to AASM guidelines [30].

### 2.5. Exercise Training Session

The interval exercise session consisted of 3-min bouts of exercise repeated 6 times at an intensity ≥ 80% of the participant’s second ventilation threshold (VT_2_), and at ≥80% of the participant’s maximum heart rate recorded during the VO_2_ max test. The rest interval was 2 min at 2.5 mph and 0 degrees of inclination. The exercise was conducted under the supervision of a certified exercise physiologist from the Canadian Society for Exercise Physiology at the CTEP lab. Each exercise session included a 5–10-min warmup, the interval workouts, and a 5-min cooldown. Continuous ECG, oxygen uptake, and ventilation were recorded with a metabolic analyzer (Vmax Encore 29, CareFusion, USA) and 12-lead ECG (CardioSoft, GE Healthcare, USA). Furthermore, the participants wore a heart rate monitor (V800, Polar Electro Oy, Finland), and the data were uploaded to the Polar Flow web service (Polar Electro Oy, Finland) for analysis. The exercise training session began around 17 h00–18 h00, when the exercise was, at most, 6 h before bedtime; this time of day has been shown to be popular for adults to exercise (US Census Bureau 2015).

### 2.6. New Sport Watch Specifications

The Polar prototype sport watch was a custom prototype of the commercial Polar M600, Polar Electro Oy, Finland. The new sport watch weighed roughly 60 g and its dimensions were 45 mm × 36 mm × 13 mm. It had a triaxial accelerometry sensor with a sampling rate of 52 Hz (STMicroelectronics, Geneva, Switzerland), a proprietary photoplethysmography sensor, and an algorithm to detect pulse-to-pulse intervals. Data were uploaded to a prototype version of a web service that used a Polar proprietary algorithm to detect a sleep period and classify it into sleep stages in 30-s epochs. The sleep variables collected and their commercial terms in brackets were sleep onset time (fell asleep time), sleep offset time (woke up time), awake (interruptions), rapid eye movement sleep stage (REM sleep), light sleep stage (light sleep), and deep sleep stage (deep sleep). The calculation of the variables is the same as the “Sleep Plus Stages” feature that is available in many Polar product series, including Grit X, Ignite, Pacer, Unite, and Vantage.

### 2.7. Statistical Analysis Process

The statistical analyses were conducted using SPSS 21.0 (IBM, Armonk, NY, USA) and SAS (9.4 version, Cary, NC, USA). All data were checked for normal distribution (Shapiro–Wilk test), sphericity, and normality of residuals before analysis was initiated. Kappa, Bland–Altman plots, sensitivity, specificity and accuracy (Equations (1)–(3)), and paired samples *t*-tests were used to compare sleep stages, total sleep time, and sleep period between PSG data scored by a sleep technician and the data recorded by the prototype sport watch:Sensitivity of sleep = TP/(TP + FN)(1)
where TP is a true positive assessment (same result obtained from the PSG and the prototype sport watch for the measurement analyzed) and FN is the false negative that represents all positive assessments in which the PSG shows the measurement analyzed but the prototype sport watch does not recognize them.
Specificity = TN/(TN + FP)(2)
where the TN is a true negative, where the PSG and the prototype sport watch agreed in recognizing that it is not the measurement analyzed, and the FP is the false positive, where the PSG does not recognize the measurement analyzed but the prototype sport watch does.
Accuracy = (TN +TP)/(TN + TP + FN + FP)(3)
where the number of correct detections from the prototype sport watch is divided by all the epoch detections.

## 3. Results

### 3.1. Participants

Participants’ characteristics are described in Table 2. Only age and the % body fat were not normally distributed. No significant differences in these characteristics were observed between the different sleep nights. The study population consisted of young Caucasian adults. VO_2_ max and grip strength showed a comparable physical capacity compared to sex- and age-matched CSEP norms [31].

### 3.2. Sleep/Awake State Detection

The sleep/awake detection outcomes are provided in Table 3, and the Bland–Altman plots are presented in Figure 2. The awake sensitivity, specificity, and accuracy are presented in Table 3. No significant differences were observed between PSG and the prototype for Total Sleep Time (*t*(35) = −0.312, *p* = 0.747 for the sleep night without exercise, and *t*(35) = −0.355, *p* = 0.725 for the sleep night with exercise) and for the period sleep time (*t*(35) = −1.688, *p* = 0.100 for the sleep night without, and *t*(35) = −0.968, *p* = 0.342 for the sleep night with exercise).

### 3.3. Sleep Stage Detection

The Bland–Altman plots representing both sleep nights for each sleep stage are provided in Figure 3. The sensitivity, specificity, and accuracy data are presented in Table 3 and the sleep stages from PSG and the prototype are presented in Table 4. No significant differences were observed between PSG and the prototype for all sleep stages for both sleep nights. Sleep night without exercise: *t*(35) = 0.312, *p* = 0.757 for the awake state; *t*(35) = −1.557, *p* = 0.127 for the REM stage; *t*(35) = 1.383, *p* = 0.174 for the light sleep stage; and *t*(35) = −0.751, *p* = 0.457 for the deep sleep stage. Sleep night with exercise: *t*(35) = 0.247, *p* = 0.80 for the awake state; *t*(35) = −1.240, *p* = 0.223 for the REM stage; *t*(35) = 0.752, *p* = 0.457 for the light sleep stage; and *t*(35) = −0.353, *p* = 0.726 for the deep sleep stage. The Kappa for the sleep night without exercise was 0.39 ± 0.13, with a range of 0.06 to 0.61 and an agreement of 60.1 ± 8.1% without a significant difference between sexes. The Kappa for the sleep night with exercise was 0.36 ± 0.16, with a range of −0.02 to 0.67 and an agreement of 59.2 ± 9.8% without a significant difference between sexes or sleep nights. No significant difference was observed between the sleep nights.

## 4. Discussion

### 4.1. Main Findings

The present study shows an overall fair agreement between the prototype sport watch (Polar Electro Oy, Finland) and the PSG sleep stages. Furthermore, no significant differences were observed between the sleep night preceded by IT exercise and the one without exercise; with performing exercise during the evening not significantly affecting the performance of the new sport watch in measuring sleep stages. This finding indicates that using the prototype sport watch with active adults, if the exercise is performed 6 h or more before bed, can be useful to support sleep hygiene and behavior interventions. Furthermore, the performance of the prototype sport watch was comparable to that of one other sleep tracking wearable.

### 4.2. Sleep Detection

Previous studies using actigraphy to detect sleep and awake states have shown generally low specificity ranges. For example, the specificity range for Fitbit devices has been reported to be between 0.20–0.61 and 0.30–0.67 for actigraphy devices [25]. The present study observed a high-performance specificity in sleep detection as compared to previous validation studies with healthy adults. The specificity for the awake state with the prototype sport watch was 0.94 ± 0.05, and the sleep state was 0.65 ± 0.21 during nights without exercise. Also, the specificity during sleep nights with exercise was 0.95 ± 0.04 for the awake state and 0.68 ± 0.22 for the sleep state. Compared to similar wearables using heart rate and wrist accelerometry, including the Garmin Forerunner 245, WHOOP (3.0), Apple Watch S6, Fitbit Charge HR, and Oura which were previously reported [10,32], the accuracy of sleep detection was similar, even slightly higher, for the prototype sport watch (~0.91 for the prototype sport watch vs. 0.90 for FitBit Charge HR vs. 0.88 for Apple Watch S6 vs. 0.89 for the Garmin Forerunner 245, vs. 0.89 for the Oura (Gen.2) and 0.86 for the WHOOP (3.0)) and the sensitivity of wake identification was higher for the prototype sport watch (~0.65) than the FitBit Charge 2 (0.61), the Apple Watch S6 (0.26), the Garmin Forerunner 245 (0.27), the Oura Gen.2 (57), and the WHOOP 3.0 (56) [10,25,32]. The prototype sport watch used for the present study shows promise for more accurate sleep time monitoring compared to similar commercially available wrist sleep trackers.

### 4.3. Sleep Stages

The sleep stage detection with wearables is often presented as either REM or NREM sleep, where stages N1, N2, and N3 are pooled [33]. Other devices like the Fitbit Charge 2 and the prototype sport watch from the present study, report the distinct stages as REM, light sleep (N1 and N2), and deep sleep (N3). Previous studies observed an overestimated time for stages N1 and N2 and an underestimated time in deep sleep with stage N3 [25]. In the present study, the Bland–Altman plots (Figure 3) showed an inversed trend; however, the sleep stage times measured by the prototype sport watch were not significantly different from the PSG recording (*p* > 0.05).

A higher accuracy was shown in the present study for the different sleep stages as compared to similar commercially available wearables using heart rate and wrist accelerometry (the Garmin Forerunner, WHOOP, Apple Watch, and FitBit Charge 2 [10,25]). The accuracy for the prototype sport watch for sleep stages ranged between 0.64 and 0.84, where light sleep stages (N1 and N2) had the lower accuracy at 0.64, followed by deep sleep at 0.81 and REM sleep at 0.84. Miller et al. [10] reported only the multi-state accuracy where the Apple Watch S6 was 0.53, the Garmin Forerunner 245 was 0.50, and the WHOOP 3.0 was 0.60. De Zambotti et al. [25] reported a better accuracy with the Fitbit Charge 2 for light sleep (0.81) but a lower accuracy for the two other sleep stages: 0.49 for deep sleep and 0.74 for REM sleep. The agreement reported by Kappa coefficients was fair to moderate (0.39 ± 0.13 without exercise and 0.36 ± 0.16 with exercise prior to bed) for the sleep stages from the prototype sport watch as compared to PSG. Moreover, the percentage agreement was around 60%. Compared to commercially available devices, as reported from Miller et al. [10], the prototype sport watch performed better than the Apple Watch S6 (0.20) and the Garmin Forerunner 245 (0.25), but it performed worse than the Oura Gen 2 (0.43) and the Whoop (0.44). These results show that as a cheaper and more user-friendly alternative to PSG, the new sport watch performs well compared to the other commercially available sleep tracker wearables and could be used to give a general idea about changes in sleep architecture and sleep quality for active people, as proposed by Miller et al. [10] and Rentz et al. [34]. Furthermore, it offers various benefits, such as the capability to monitor sleep in a free-living environment, potentially providing a more accurate representation of an individual’s typical sleep patterns [34].

### 4.4. Limitations

The present study was conducted with attention to details such as the inclusion of a familiarization night, during which the participant had a night of sleep with the equipment prior to the experimental periods. However, some limitations need to be considered. The sample included only healthy Caucasian adult participants, in whom there was minimal variation in age, skin color, and health issues that can add variations and/or difficulties for the sensors to obtain data [7]. For individuals with sleep disorders, new algorithms and sensors are probably needed and were not included in this study. Furthermore, the device used for this study was a prototype from Polar (Polar Electro Oy Finland) but was of similar size and appearance to the commercially available products (Polar Electro Oy, Finland) and the calculation of the sleep variables was obtained with the Polar proprietary algorithm that is in use in several currently available products. However, as the algorithms are not publicly accessible and are restricted from modification by anyone other than the manufacturer, this hampers the potential to incorporate new clinical algorithms tailored for complex sleep conditions.

### 4.5. Recommendations and Applications

The general public is more informed on sleep health and the impact of lack of sleep, which probably explains the rapid growth in the market of wearables measuring sleep within the past decade [33]. However, few of these devices have been validated against the gold standard for sleep assessment, PSG. The awareness of sleep health and sleep hygiene that can be achieved by using wearable, sleep-monitoring devices could help individuals to change some sleep habits through the monitoring of their sleep quality [17]. Furthermore, athletes that are part of the target population often confront challenges in securing adequate sleep due to their demanding competition and travel schedules, along with rigorous training regimens that can disrupt their rest. While prioritizing sleep remains crucial, some individuals struggle with initiating and maintaining it [5]. In such cases, athletes can benefit from enhancing their sleep hygiene practices. Developing healthy sleep patterns can be achieved through consistent routines and creating an optimal sleep environment with the help of wearables to monitor their sleep. Vitale et al. [5] provide examples of effective sleep hygiene measures, including maintaining regular wake-up times, establishing nightly routines, and minimizing stimulants and distractions. While it may not be feasible for athletes to adopt all these recommendations, they are encouraged to incorporate as many as possible to maximize their sleep quality and overall well-being. However, sleep specialists and industry will need to collaborate closely to improve the validity of devices to detect sleep stages and sleep disorders regardless of the population. Wulterkens et al. [35] seem to show a promising algorithm based on wrist-worn photoplethysmography and accelerometry that shows a Cohen’s kappa of 0.62 and an accuracy of 76.4% with patients with a broad variety of sleep disorders, and it could probably be implemented in the next generation of sport watches. If this addition can be validated, the potential of using multi-sensor wearables will be seen not only in research and clinical settings but also in ecologically valid in-home study environments. These devices offer the capability of monitoring sleep, which could potentially be more cost-effective for healthcare companies compared to expensive clinical settings like those proposed by Roberts et al. [36].

### 4.6. Conclusions

The new prototype sport watch (Polar Electro Oy, Finland) shows an equal or better accuracy and agreement with PSG in sleep-stage recognition, as compared to other commercially available devices in the free-living environment. Furthermore, no significant difference was observed between sleep nights with and without exercise before bedtime, showing that it can be used with active adults. More investigation will be needed to validate the prototype sport watch in populations other than healthy adults.

## Figures and Tables

**Figure 1 sensors-24-02218-f001:**
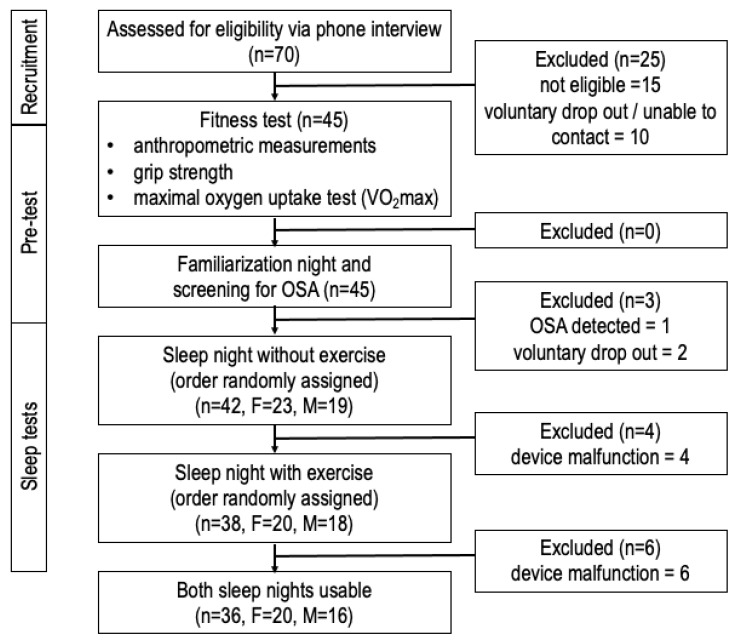
Method flow chart.

**Figure 2 sensors-24-02218-f002:**
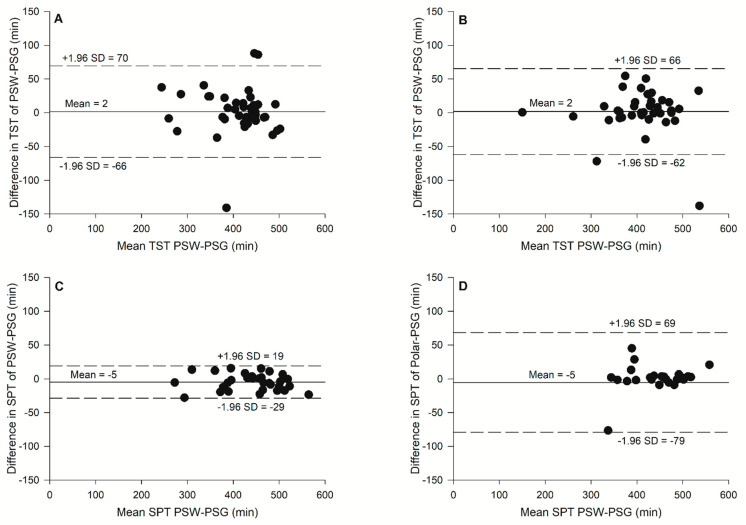
Bland-Altman plots for (**A**) total sleep time (TST) without exercise, (**B**) total sleep time (TST) with exercise, (**C**) sleep period time (SPT) without exercise, (**D**) sleep period time (SPT) with exercise against the mean of the two measurements with the prototype sport watch (PSW) and the gold standard, PSG.

**Figure 3 sensors-24-02218-f003:**
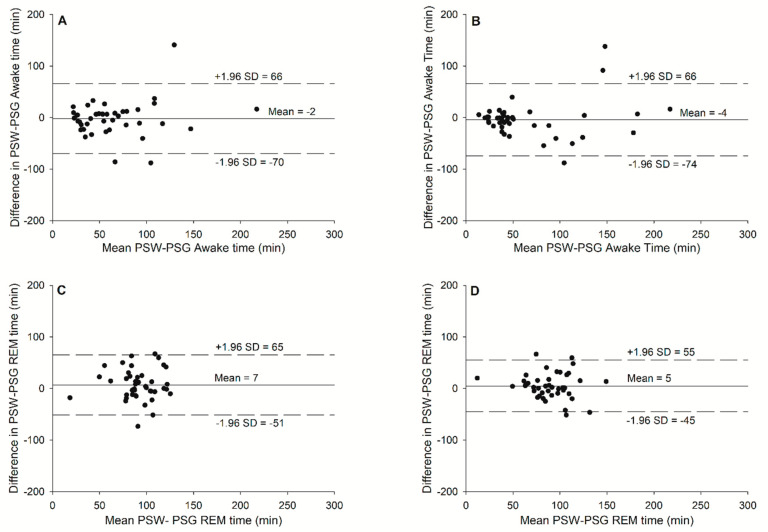
Bland-Altman plots for (**A**) awake time without exercise, (**B**) awake time with exercise, (**C**) REM stage time without exercise, (**D**) REM stage time with exercise, (**E**) N1 and N2 (light sleep stage) time without exercise, (**F**) N1 and N2 (light sleep stage) time with exercise, (**G**) N3 (deep sleep stage) time without exercise, and (**H**) N3 (deep sleep stage) time with exercise against the mean of the two measurements with the prototype sport watch (PSW) and the gold standard, PSG.

**Table 1 sensors-24-02218-t001:** Non-exhaustive comparison of different commercial wearable sleep-tracking devices.

Devices	Outcomes		Device’s Placement of Photoplethysmography Light Source	
	Sleep Data	Measurements	Wrist	Other
Oura Ring	Sleep time Wake time Light sleep Deep sleep REM sleep	Heart rate Heart rate variability Respiratory rate Body temperature Night-time movement		Ring finger Infrared
Fitbit Charge 2	Sleep time Wake time Light sleep Deep sleep REM sleep	Heart rate Heart rate variability Night-time movement	Wrist Infrared Green LED Red LED	
Apple Watch	Sleep time Wake time Light sleep Deep sleep	Heart rate Night-time movement	Wrist Infrared Green LED Red LED	
Garmin Forerunner 945	Sleep time Wake time Light sleep Deep sleep REM sleep	Heart rate Saturation? Night-time movement	Wrist Green LED	
WHOOP (3.0)	Sleep time Wake time Light sleep Deep sleep REM sleep	Skin temperature Heart rate Night-time movement	Wrist Green LED	

HRV: heart rate variability, EBE analysis: epoch by epoch analysis of sensitivity, specificity, and overall agreement to compare performance to differentiate sleep. References from: Miller et al. (2022), Lugan et al. (2021), Rentz et al. (2021), and De Zambotti et al. (2018).

**Table 2 sensors-24-02218-t002:** Participants’ physiological characteristics from usable data during sleep night.

	Sleep Night without Exercise			Sleep Night with Exercise			Both Sleep Nights		
	Females *n* = 23	Males *n* = 19	All *n* = 42	Females *n* = 20	Males *n* = 18	All *n* = 38	Females *n* = 20	Males *n* = 16	All *n* = 36
Age (years)	26 (4)	27 (5)	27 (5)	26 (4)	27 (5)	27 (4)	26 (4)	27 (5)	27 (4)
BMI (kg∙m^−2^)	23.4 (3.5)	24.2 (2.7)	23.9 (3.3)	23.5 (3.8)	24.1 (2.6)	23.7 (3.2)	23.5 (3.7)	24.0 (2.6)	23.6 (3.3)
% body fat (%)	29.7(6.5)	20.6 (5.6) *	25.9 (7.7)	29.7 (6.5)	20.6 (5.6) *	25.5 (7.8)	29.8 (6.7)	20.2 (5.9) *	25.3 (7.9)
WHR	0.80 (0.04)	0.86 (0.04) *	0.83 (0.06)	0.80 (0.04)	0.86 (0.06) *	0.83 (0.05)	0.80 (0.04)	0.85 (0.05) *	0.82 (0.05)
Grip Strength (kg)	64.6 (18.6)	105.2 (18.4) *	81.8 (27.2)	63.8 (19.7)	101.9 (16.1) *	80.7 (26.2)	63.8 (20.0)	102.1 (16.6) *	80.1 (26.9)
VO_2_max (mL∙kg^−1^∙min^−1^)	42.0 (7.2)	52.4 (7.7) *	46.2 (9.0)	42.0 (7.2)	53.0 (7.6) *	46.9 (9.4)	42.1 (7.7)	52.7 (7.8) *	46.8 (9.4)
VTh2 (mL∙kg^−1^∙min^−1^)	37.3 (6.4)	44.0 (7.2)*	40.0 (7.5)	37.6 (6.7)	43.6 (7.6) *	40.3 (7.6)	37.6 (6.7)	43.4 (7.8) *	40.3 (7.7)

Mean (SD); BMI, body mass index; WHR, waist–hip ratio; VO_2_ max, maximal oxygen uptake; VTh2, ventilator threshold 2; * *p* ≤ 0.05 between sex.

**Table 3 sensors-24-02218-t003:** Sensitivity, specificity, and accuracy for the different sleep stages for the sleep night without and with exercise.

	Females (*n* = 23)		Males (*n* = 19)		All (*n* = 42)	
	Mean (SD)	Range	Mean (SD)	Range	Mean (SD)	Range
Sleep without Exercise						
Sensitivity						
Awake	0.64 (0.22)	0.04–0.87	0.67 (0.02)	0.24–0.97	0.65 (0.21)	0.04–0.97
REM	0.57 (0.15)	0.24–0.83	0.57 (0.17)	0.28–0.83	0.57 (0.16)	0.24–0.83
N1 and N2	0.67 (0.09)	0.54–0.82	0.64 (0.09)	0.43–0.81	0.66 (0.09)	0.43–0.82
N3	0.48 (0.20)	0.00–0.75	0.35 (0.21)	0.00–0.65	0.42 (0.21)	0.00–0.75
Specificity						
Awake	0.95 (0.03)	0.87–1.00	0.93 (0.07)	0.80–0.99	0.94 (0.05)	0.80–1.00
REM	0.93 (0.04)	0.85–0.99	0.90 (0.06)	0.79–1.00	0.91 (0.05)	0.79–1.00
N1 and N2	0.63 (0.08)	0.39–0.78	0.63 (0.09)	0.47–0.80	0.63 (0.09)	0.39–0.81
N3	0.88 (0.06)	0.75–0.96	0.90 (0.07)	0.71–1.00	0.89 (0.07)	0.71–1.00
Accuracy						
Awake	0.93 (0.03)	0.85–0.98	0.89 (0.08)	0.68–0.96	0.91 (0.06)	0.68–0.98
REM	0.85 (0.05)	0.73–0.92	0.84 (0.06)	0.73–0.92	0.84 (0.05)	0.73–0.92
N1 and N2	0.65 (0.06)	0.49–0.72	0.64 (0.07)	0.49–0.74	0.64 (0.06)	0.49–0.74
N3	0.81 (0.05)	0.67–0.88	0.81 (0.06)	0.68–0.91	0.81 (0.06)	0.67–0.91
Sensitivity						
Sleep with Exercise						
Awake	0.69 (0.21)	0.19–0.95	0.66 (0.03)	0.21–0.98	0.68 (0.22)	0.19–0.98
REM	0.59 (0.18)	0.18–0.85	0.49 (0.19)	0.10–0.74	0.54 (0.19)	0.09–0.85
N1 and N2	0.64 (0.12)	0.40–0.84	0.67 (0.09)	0.49–0.82	0.66 (0.11)	0.40–0.84
N3	0.45 (0.21)	0.00–0.73	0.36 (0.19)	0.00–0.63	0.41 (0.20)	0.00–0.73
Specificity						
Awake	0.95 (0.04)	0.85–1.00	0.94 (0.04)	0.86–0.99	0.95 (0.04)	0.85–1.00
REM	0.90 (0.05)	0.77–0.97	0.91 (0.06)	0.79–1.00	0.90 (0.05)	0.77–1.00
N1 and N2	0.64 (0.11)	0.39–0.82	0.57 (0.09)	0.41–0.75	0.61 (0.11)	0.39–0.82
N3	0.87 (0.07)	0.68–0.96	0.90 (0.06)	0.78–1.00	0.88 (0.07)	0.68–1.00
Accuracy						
Awake	0.93 (0.05)	0.73–0.98	0.90 (0.06)	0.72–0.96	0.92 (0.06)	0.72–0.98
REM	0.84 (0.07)	0.64–0.93	0.82 (0.07)	0.69–0.90	0.83 (0.07)	0.64–0.93
N1 and N2	0.64 (0.09)	0.47–0.79	0.62 (0.07)	0.49–0.75	0.63 (0.08)	0.47–0.79
N3	0.81 (0.06)	0.66–0.91	0.80 (0.07)	0.66–0.91	0.80 (0.07)	0.66–0.91

Mean (SD); REM, rapid eye movement sleep stage; NI and N2: sleep stage 1 and 2 (light sleep stages); N3: sleep stage 3 (deep sleep stage).

**Table 4 sensors-24-02218-t004:** Sleep metrics values from PSG and the prototype.

	PSG	Prototype
	Mean (SD)	Mean (SD)
Without Exercise		
SPT (min)	450 (70)	446 (66)
TST (min)	411 (65)	412 (63)
SOn (min)	88 (42)	94 (45)
SOff (min)	522 (60)	519 (61)
Awake (nb epoch)	115 (63)	119 (80)
REM (nb epoch)	176 (52)	189 (50)
N1 and N2 (nb epoch)	483 (108)	457 (103)
N3 (nb epoch)	160 (58)	170 (57)
With Exercise		
SPT (min)	454 (74)	451 (57)
TST (min)	408 (75)	410 (73)
SOn (min)	109 (92)	109 (100)
SOff (min)	519 (124)	514 (118)
Awake (nb epoch)	117 (78)	116 (99)
REM (nb epoch)	180 (49)	188 (46)
N1 and N2 (nb epoch)	498 (105)	488 (94)
N3 (nb epoch)	153 (59)	156 (46)

SPT, sleep period time; TST, total sleep time; SOn, sleep onset recording time; SOff, sleep offset recording time; nb, number; REM, rapid eye movement sleep stage; NI and N2, sleep stage 1 and 2 (light sleep stages); N3, sleep stage 3 (deep sleep stage).

## Data Availability

The original contributions presented in the study are included in the article, further inquiries can be directed to the corresponding author.

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
