# Peer review of "Performance Evaluation of a New Sport Watch in Sleep Tracking: A Comparison against Overnight Polysomnography in Young Adults"

_sensors, 2024, doi:10.3390/s24072218_

Round 1
Reviewer 1 Report
Comments and Suggestions for Authors
The work presented by Parent et al., describes a clinical experiment of evaluating a wrist worn technology aimed at collecting sleep parameters, including sleep architecture, and establishing concordance with a conventional benchmark, polysomnography, used in both sleep medicine clinical practice and clinical research. The conflict of interest has an appropriate disclosure that the study was partially funded by the technology manufacturer.
While this publication has certain merits, I have a number of major comments that need to be seriously considered before this research can be considered in the Sensors Journal.
11. The authors made a big shortcut connecting actigraphy, polysomnography, and different sensor output used in commercial fitness trackers to measure sleep parameters. Because of this shortcut, the introduction section of the manuscript is confusing at best. The authors should keep in mind that the Sensors Journal has a very broad audience, and the definitions and features of the sleep data collection methods need to be explained thoroughly which would improve readability of the manuscript and make the content of methods, results, and discussion a lot clearer. For example, there is no description of what sensors and measures are used for polysomnography until one gets to the method section which explains at home polysomnography. It is very important to highlight that polysomnography uses multiple types of sensors and the most important of them is EEG which collects brain waves. There is no mention of this very important component in the introduction. The description of actigraphy is very confusing. Historically, actigraphy is not a device; it is a data collection method that uses output of an accelerometer to determine periods of sleep and wakefulness. Later, wrist worn actigraphy devices started being used also for quantifying parameters of physical activity. The limitations of actigraphy are well-known and should be highlighted in the introduction. The technology under consideration in this manuscript is a combination of some sort of accelerometer or IMU that has a combination of sensors and, apparently, also include a PPG sensor which collects vital sign data, such as heart rate and heart rate variability. The combination of accelerometry and vital sign data is not actigraphy.
2. The problem the authors are trying to address is to substitute at home polysomnography with a wrist worn technology that can collect sleep parameters, including sleep architecture, with reasonable sensitivity, specificity, and accuracy. The proposed solution is a combination of standard actigraphy with some vital sign data that is being compared to polysomnography. It would be very helpful if the authors summarize peer-reviewed literature in a table format of similar technologies in comparison with polysomnography with appropriate references. Such a summary, as a part of the introduction, will help the reader to understand the rationale for this study and what problem that this study is addressing. Many fitness trackers, marketed directly to consumers, use a combination of actigraphy and vital sign data to provide information to the users about their sleep and quantify their exercise. However, many technologies used in the fitness trackers including both sensors and data processing algorithms, are proprietary to the manufacturers, making their use for clinical research purposes very limited.
23. Lines 21 – 23. The authors should be very careful about making claims of the utility of the technology under investigation. They state: “sleep quality measurements could provide an opportunity for early intervention, thus preventing negative health consequences of sleep disturbances before they arise”. What does this mean in clinical practice? Should patients wear a fitness tracker and bring it to their physicians to review their sleep data? Should these data be ubiquitously collected similar to laboratory tests? Why?
44. The information about the technology under investigation is sorely missing. It’s hardly any information at all about this wrist worn tracker that the whole study is dedicated to. I strongly recommend that the authors follow the best practice in the scientific community to describe features and properties of technology under investigation as recommended by Manta at all, 2021 https://pubmed.ncbi.nlm.nih.gov/34179682/. For example, it is not clear what is the status of that technology under investigation? Is it a commercial fitness tracker, a prototype of a medical device, or something else? No information about variables collected and data processing software. Without this information, it is impossible to reproduce the results of the experiments described in this paper.
45. The authors are very inconsistent about terminology. They called that technology under investigation a wearable. Unfortunately, this term is being used in the scientific literature though the definition for this term, as far as I know, doesn’t exist. I recommend that the authors use definitions that are consistent with regulatory science or are accepted in the scientific community with definitions that can be referenced in the peer-reviewed literature. The examples are digital health technologies (DHT) as defined by the FDA or biometric monitoring technologies as defined by Digital medicine Society. The authors should proofread the manuscript for consistency and use the same term everywhere. Some sections of the discussion refer to the technology under investigation as new device without even mentioning the name which is unacceptable. The term “device” should not be used unless the technology under investigation has a designation of a medical device. If this is the case, the regulatory clearance/approval information should be referenced in the manuscript.
56. Lines 157 – 158. Which exactly sleep data were collected? The authors should define very clearly the variables collected and defined them. Some of the definitions provided in the manuscript outside of the standard actigraphy data definitions. What is the sleep period? How different is sleep period from total sleep time? I couldn’t find a definition of the sleep period anywhere and, yet, this is one of key parameters.
67. Table 3. What is nb epoch? It’s not defined anywhere.
78. Lines 212-219. Comparing the results described in this paper with the results obtained by Actiwatch as described by Marino et al. is a mistake. The Actiwatch device is a medical actigraphy device that doesn’t collect any vital sign data. If the authors want to compare the technology under investigation with Actiwatch, it should be done correctly, highlighting appropriate caveats. The same applies to comparison of the technology under investigation with smart phones that are not designed to collect any vital sign data.
87. The title of the manuscript is misleading. This is an evaluation study, not validation. To make a validation claim, the acceptance parameters need to be established a priori to define under which conditions a validation claim is valid and under which condition is not.
98. Lines 259 – 261. The statement about additional validation needed for conditions such as insomnia or sleep apnea is erroneous. The authors should know that data processing algorithms, used for actigraphy, make certain assumptions, including a major period of rest for a certain duration. If these assumptions are violated for certain medical conditions, the algorithms are invalid. New algorithms need to be developed that can capture sleep parameters for certain sleep disorders that take into account sleep patterns under these conditions, e.g. intermittent sleep, no major sleep period, altered sleep architecture, etc. This is where commercial fitness trackers developed for general populations are useless because their data processing algorithms are not publicly available and cannot be modified by anyone other than the manufacturer.
Author Response
Please find attached our replies to the reviewers' comments.

Reviewer 2 Report
Comments and Suggestions for Authors
1. The content of 1.1 in the introduction can be combined with the previous paragraph for easy reading.
2. It is recommended to detail in Materials and Method how the prototype device monitors sleep stages,sleep period time, and total sleep time.
3. Table2, It is suggested to distinguish "sleep without exercise" and "sleep with exercise" by adding auxiliary lines to make the data more convenient to read.
4. Line180-181, “No significant differences were observed between PSG and the prototype for all sleep stages for both sleep night”It is better to change “night”to “nights”.
5. Line213-214, “Compared to the Actiwatch AW-64 (Minimitter, Inc, Bend, OR) reported by Marino et al., the accuracy of sleep detection was similar, even slightly higher.”It is recommended to change to an updated report to make device effectiveness more convincing.
6. The references are old, and it is suggested to add more recent reports.
Comments on the Quality of English Language
Line180-181,It is better to change “night”to “nights”.
Author Response

(The authors gave the same response as above.)

Reviewer 3 Report
Comments and Suggestions for Authors
I sincerely appreciate your dedicated efforts and the valuable contributions stemming from your research. In relation to this paper, I would like to ask some questions:
1. Additional explanation should be needed for the algorithm mentioned in line 61.
2. There seems to be an error in the time axis of figure 2, (A) & (B). The total time should necessarily be greater than the period time, as indicated in table 3.
3. It seems that the overall average value for the specificity of N3 in "sleep without exercise" in table 2 has been miscalculated. It is lower than the respective averages for ‘Females’ and ‘Males'.
Author Response

(The authors gave the same response as above.)

Round 2
Reviewer 3 Report
Comments and Suggestions for Authors
The author revised the manuscript appropriately in consideration of the reviewer's questions. In addition, author added material in the introduction to help reader's understand. Therefore, this manuscript is appropriate to be published.